# Field Application of a Vis/NIR Hyperspectral Imaging System for Nondestructive Evaluation of Physicochemical Properties in ‘Madoka’ Peaches

**DOI:** 10.3390/plants11172327

**Published:** 2022-09-05

**Authors:** Kyeong Eun Jang, Geonwoo Kim, Mi Hee Shin, Jung Gun Cho, Jae Hoon Jeong, Seul Ki Lee, Dongyoung Kang, Jin Gook Kim

**Affiliations:** 1Division of Applied Life Science, Graduate School of Gyeongsang National University, 501, Jinju-daero, Jinju-si, Gyeongsangnam-do 52828, Korea; 2Department of Bio-industrial Machinery Engineering, College of Agriculture and Life Science, Gyeongsang National University, 501, Jinju-daero, Jinju-si, Gyeongsangnam-do 52828, Korea; 3Institute of Agriculture and Life Sciences, Gyeongsang National University, 501, Jinju-daero, Jinju-si, Gyeongsangnam-do 52828, Korea; 4Fruit Research Division, National Institute of Horticultural and Herbal Science, Wanju 55365, Korea; 5Department of Horticulture, College of Agriculture and Life Science, Gyeongsang National University, 501, Jinju-daero, Jinju-si, Gyeongsangnam-do 52828, Korea

**Keywords:** fruit quality, quality prediction, plant phenotyping, orchard management

## Abstract

Extensive research has been performed on the in-field nondestructive evaluation (NDE) of the physicochemical properties of ‘Madoka’ peaches, such as chromaticity (a*), soluble solids content (SSC), firmness, and titratable acidity (TA) content. To accomplish this, a snapshot-based hyperspectral imaging (HSI) approach for filed application was conducted in the visible and near-infrared (Vis/NIR) region. The hyperspectral images of ‘Madoka’ samples were captured and combined with commercial HSI analysis software, and then the physicochemical properties of the ‘Madoka’ samples were predicted. To verify the performance of the field-based HSI application, a lab-based HSI application was also conducted, and their coefficient of determination values (R^2^) were compared. Finally, pixel-based chemical images were produced to interpret the dynamic changes of the physicochemical properties in ‘Madoka’ peach. Consequently, the a* values and SSC content shows statistically significant R^2^ values (0.84). On the other hand, the firmness and TA content shows relatively lower accuracy (R^2^ = 0.6 to 0.7). Then, the resultant chemical images of the a* values and SSC content were created and could represent their different levels using grey scale gradation. This indicates that the HSI system with integrated HSI software used in this work has promising potential as an in-field NDE for analyzing the physicochemical properties in ‘Madoka’ peaches.

## 1. Introduction

Hyperspectral imaging (HSI) has been intensively used as an effective tool for plant phenotyping in the nondestructive evaluation (NDE) fields [1], agricultural products [2], food safety, and quality [3] due to its rapid and accurate detection and classification of target materials. Moreover, HSI can simultaneously provide the spectral and spatial information of a target material by combining spectroscopic analysis and image processing [4,5,6]. It, therefore, has a high potential to be used for online sorting application systems of various foods [5]. Consequently, this allows users to detect or classify a target material through pixel-based chemical images from agricultural products [7,8,9].

Despite the great advantages of HSI technologies for agricultural NDE, the in-field-based HSI technologies for agricultural NDE still have some critical issues. The total cost of an in-field based HSI system is expensive, and it is also difficult for users to handle the HSI system due to its complexity, such as in the scanning method (line or snapshot scanning), long image acquisition time (more than regular machine vision), sensitive control for line scanning, low pixel resolution, large amount of sample information, etc. [10].

Moreover, to build robust prediction models based on various chemometric methods, spectral calibration is essential for establishing high accuracy [8,11]. However, because the spectral calibration is strongly affected by weather conditions such as brightness and the movement of clouds, the majority of detection and classification chemometric models have been developed under a precisely controlled environment [6,12,13].

To overcome these difficulties, commercial companies such as Perception Park (Graz, Austria), Prediktera (Umeå, Sweden), and perClass BV (Delft, Netherlands) have released their commercial hyperspectral imaging software, which allows researchers and agricultural entrepreneurs to analyze hyperspectral images more easily. They have been attempting to use this software in a wide variety of industrial [14,15], medical [16], and agricultural fields [17,18]. In the agricultural field, the HSI system has been widely applied to fruit ripeness, soluble solids content (SCC), firmness, bruising, fungal contamination, nutrients, and so on [10,19,20,21,22,23]. Therefore, in this study, we conducted an NDE of fruit quality using a commercial hand-held visible near-infrared (Vis/NIR) hyperspectral camera (Specim IQ, Specim, Oulu, Finland) and classification software (perClass Mira, Delft, Nederlands) which performs an automatic selection of machine learning models with various preprocessing methods for field applications.

Peaches (*Prunus persica* (L.) Batsch) are one of the most widely consumed fruits, which are cultivated in both hemispheres because of their unique flavor and high nutrition [24,25]. Peaches can be divided into melting (< 8 N firmness) and non-melting (>16 N firmness) types depending on how much of the flesh softens as they ripen [26,27]. Moreover, peach fruits are vulnerable to climate change because they quickly ripen and lose their quality even at room temperature [28]. Thus, ripeness is an important factor for fresh peaches during market distribution.

As a result of global warming, the average yearly temperature might rise by 0.3 to 4.8 degrees. The worldwide mean annual temperature will rise by 0.3 to 4.8 °C by 2100, and many agricultural crops will likely experience temperatures that are higher than the global average [29]. Hence, predicting the quality parameters of peaches has recently attracted substantial attention in the fields of agricultural NDE because of their temperature sensitivity [23,24,30].

Approximately 30 kinds of peach cultivars, including ‘Madoka’ peaches, have been cultivated in the Republic of Korea. ‘Madoka’ is the typical melting type that quickly softens during its ripening stage [25]; therefore, we have selected ‘Madoka’ peaches as our target in field applications due to its physiological response to temperature conditions.

In a lab-based environment, important physicochemical properties of agricultural products could be measured and predicted under light source, controlled scanning speed, and minimized vibration [3,19,20]. However, field-based HSI for agricultural products has some critical issues, such as irregular intensity of sunlight, the movement of clouds, dirt on the surface of fruits, and so on. They can affect the brightness of the obtained hyperspectral images and decrease the overall HSI performance. Accordingly, improving the performance of the field-based HSI technologies is an emerging issue for the NDE of agricultural products [20].

Therefore, the main objective of this study was to evaluate the physicochemical properties of ‘Madoka’ peaches. Their chromaticity, soluble solids content (SSC), firmness, and titratable acidity (TA) were evaluated using a commercial hyperspectral camera and software for field application of a commercial HSI system. To verify its accuracy, the predicted values produced by the commercial software were compared with their measured values by a colorimeter, refractometer, firmness meter, and pH meter. In addition, the accuracy of the predicted values was acquired from laboratory (indoor) and field (outdoor) studies to analyze the differences and accuracies between the two environments. Our detailed objectives are summarized as follows:Establish two Vis/NIR HSI systems (indoor and outdoor) for obtaining hyperspectral data of ‘Madoka’ peaches, with an increase in their growth period,Measure the physicochemical properties of ‘Madoka’ peaches in an indoor and outdoor environment and compare their predicted values with those of the commercial software,Analyze the predicted results from both the indoor and outdoor environment and determine a preprocessing method which produces the highest accuracy among various preprocessing methods,Demonstrate feasibility by providing pixel-based visualization of physicochemical distribution created by hyperspectral images taken from an indoor and outdoor environment.

## 2. Results

### 2.1. Spectra Extraction First Item

Figure 1 shows the average spectra acquired from the ROI regions in both lab- and field-based hyperspectral sample images, ranging from 400 to 1000 nm. Both spectra were divided into six groups according to each harvesting date. The obtained spectra show a high absorption valley appearing at the 680 nm and 970 nm regions in both spectra plots. It was found that there was no major difference between the two methods.

### 2.2. Measured Physicochemical Properties

Table 1 shows the measured physicochemical properties of peach ‘Madoka’. The ranges of chromaticity (a*), SSC, firmness, and TA were shown, and their average, standard error (SE), standard deviation (STDEV) values were represented, respectively.

### 2.3. Chromaticity (a*) Prediction

The result of predicting chromaticity (a*) in ‘Madoka’ is summarized in Table 2. As shown in the table, under lab-based HSI, SGD 1st derivative differentiation shows the highest R^2^ value (R^2^ = 0.87 at the validation set) in both the calibration and validation set with the lowest bias values. On the other hand, under field-based HSI, smoothing has the highest accuracy (R^2^ = 0.85 at the validation set).

### 2.4. SSC Prediction

The results of the SSC content analysis of ‘Madoka’ peaches are summarized in Table 3. As shown in the table, SGD 1st derivative differentiation shows the highest (R^2^ = 0.87) accuracy under lab-based HSI application. In addition, smoothing and SGD 2nd derivative differentiation show good performance (R^2^ = 0.89 and 0.85) under field-based HSI.

### 2.5. Firmness Prediction

The firmness prediction results of ‘Madoka’ are represented in Table 4. As can be seen, it was observed that the accuracies of the calibration set were relatively much lower than those of the validation set. SGD 2nd derivative differentiation shows the best accuracies (R^2^ = 66) under the lab-based HSI application. In the field-based HSI, SGD 1st derivative differentiation shows the highest accuracies (R^2^ = 0.68) under field-based HSI.

### 2.6. TA Prediction

Table 5 shows the results of the TA contents in ‘Madoka’. In the lab-based HSI, R^2^ values of the calibration and validation set were between 0.5 and 0.7. In the field-based HSI, the model accuracy was about 0.7, which is a slightly higher value than that of the lab-based HSI outcome.

### 2.7. Visualization of Physicochemical Properties

The pixel-based chemical imaging of the physicochemical component distribution in ‘Madoka’ was created by the prediction model chosen by perClass Mira software. Figure 2 shows the representative resultant hyperspectral images with an increase in SSC contents and chromaticity (a*) values. The measured values of a* and SSC contents were acquired from the ‘Madoka’ samples harvested between 13 July and 3 August. The resultant images for firmness and TA content distribution were not shown due to their low R^2^ values. In Figure 2, the resultant hyperspectral images were described by a linear gray scale with different intensities being used for each pixel and compared with their RGB images, facilitating improved understanding of the spatial variation in chromaticity (a*) and SSC in the ‘Madoka’ samples.

## 3. Discussion

### 3.1. Spectral Analysis

Spectral analysis of a target material can provide material information about its chemical composition and physical properties. Every agricultural product includes nearly the same constituents contributing to the absorption (or reflection) spectra, such as chlorophyll, carotenoids, water, proteins, waxes, starches, structural biochemical molecules, and so on [31]. Therefore, these components respond to electromagnetic radiation as a function of wavelengths to their composition and physical properties.

The typical shape and absorption peaks (675 nm and 970 nm) of the ‘Madoka’ spectra were acquired, as shown in Figure 1 [19,23,32]. The acquired sample spectra presented similar spectral trends with different intensities. The absorption peaks at 675 nm and 970 nm are related to chlorophyll a and the second overtone stretching of O-H vibrations of water and sugar components [19,23]. Increasing the time until harvest leads to the maturity of yellow ‘Madoka’. This decreases the chlorophyll content and can be used to evaluate the maturity of fresh peach fruits [33]. From the spectral data in both lab- and field-based HSI, it was difficult to determinate the criteria for high quality ‘Madoka’ as well as distinguish the critical differences between the two methods.

### 3.2. Model Evaluation Factors

The model accuracy was evaluated by conducting a quantitative analysis based on four preprocessing methods with evaluation factors such as the coefficient of determination (R^2^), standard error of prediction (SEP), residual prediction deviation (RPD), root mean square error of cross validation (RMSECV), and bias values. These evaluation factors have been commonly used for cross-validation and robust model establishment [4,6,34].

SEP indicates the prediction ability of the used model, and RPD is the ratio of the standard deviation of the validation set divided by the root mean square errors of prediction (RMSEP), describing by which component its prediction accuracy has been affected compared to the mean composition for the total samples [34]. RPD can also assess NIR calibration performance. In general, an excellent prediction model can be evaluated by the following criteria: high R^2^, high RPD, low bias, low SEP, and low RMSECV [35,36].

These factors are intensively used to evaluate the performance of chemometric methods such as partial least squares-discriminant analysis (PLS-DA), principal component analysis (PCA), least-squares support-vector machine (LS-SVM), and so on [5,36,37]. Among them, the PLS based chemometric technique with the four preprocessing methods may be automatically selected by perClass Mira software because they have been intensively applied to NDE applications for assessing fresh peach quality with high accuracy [1,19,20,21,22,23,38,39]. Therefore, an appropriate model was selected for the NDE of the physicochemical properties of ‘Madoka’ peaches, and their pixel-based chemical images were created.

### 3.3. Analysis of Model Prediction

The model prediction accuracies of the physicochemical properties in ‘Madoka’ are summarized in Table 2, Table 3, Table 4 and Table 5. Each preprocessing method includes the calculated values of the above evaluation factors under lab- and field-based HSI applications. Besides the preprocessing methods used here, other methods, such as normalization, multiplicative scatter correction, and standard normal variate, can be utilized to develop chemometric models to improve model performance. However, only four preprocessing methods could be selected by the software used.

The overall values of the firmness and TA indices gradually decrease as the time until harvesting increases; however, chromaticity (a*) and SSC content values show an opposite trend [33,40]. This color change on the peaches’ surface is consistent with their decreasing chlorophyll a content and increasing anthocyanin components [41]. Therefore, the a* component can describe the dramatic maturity stage change in red colored fruits changing from green (negative) to red (positive) color [42]. As shown in Table 2, the accuracies of the a* values of lab- and field-based HSI results were nearly identical and statistically significant (R^2^ = 0.84 to 0.87).

Conversely, the firmness and TA prediction exhibited a major difference in R^2^ values between the calibration (R^2^ = 0.6 to 0.7) and validation set (R^2^ = 0.7 to 0.85). Moreover, their prediction accuracies were much lower than chromaticity (a*) and SSC prediction. This indicates that the performance of firmness and TA contents, predicted by the automatically selected model, was statistically non-significant. In the current study, a single model was developed and applied to simultaneous identification of the physicochemical qualities of ‘Madoka’ for rapid and real-time field application using a commercial HSI camera and software. Based on the above analysis, we concluded that the developed model cannot perform multiple quality factors; thus, a different approach for developing optimal models for each quality factor is needed for the simultaneous prediction of ‘Madoka’ qualities.

### 3.4. Visualization of Physicochemical Properties

The conventional spectroscopy technique can create one set of spectral data per individual target. However, hyperspectral images can produce full spectral data from every pixel of the target image. This is the critical benefit of HSI and can provide an interpretation of the particular chemical components in a specific pixel of the target image. Therefore, the sample images obtained under two different environments can show different spatial and spectral information from their pixel data.

As shown in Figure 2, the RGB images were used for comparison with their hyperspectral images, and the resultant hyperspectral images present different a* values and SSC on a linear grey scale with different intensities being used for each pixel. This facilitates improved interpretation of the spatial variation in a* values and SSC in ‘Madoka’ samples. The differences of the a* values and SSC are hardly distinguished by the naked eye, except for yellow ‘Madoka’ samples. However, the developed model could clearly classify the different a* and SSC values based on the number of black and white pixels, which increase with their levels. As seen in Figure 2a, it is particularly difficult to distinguish the peach’s maturity in an RGB image with the naked eye, except for an a* value of –2.98, because red and pink colors are irregularly mixed with the yellow regions of the RGB sample images. However, the developed model detects and differentiates the increasing a* values with a grey scale. This can be also observed in the SSC (see Figure 2c and d) from 12.3 to 21.9%.

Although the a* values and SSC contents and their different levels could be detected, regular intervals of a* values and SSC could not be achieved in this study due to their fast ripening characteristics. According to the measured SSC values, it seems that the ‘Madoka’ samples used ripened between 23 and 27 July. Therefore, the hyperspectral images for gradual variation in peach maturity could not be obtained with the ‘Madoka’ samples used. Moreover, the dramatic change in the grey scale can be observed in Figure 2c; thus, color difference between the two groups reveals the SSC level. Consequently, the field application of the commercial HSI camera and software used in this study has considerable potential for the NDE of chromaticity and SSC in ‘Madoka’ peaches.

## 4. Materials and Methods

### 4.1. Sample Preparation

The ‘Madoka’ peaches utilized in this study were grown at an orchard located in Jiphyeon-myeon, Jinju-si, Gyeongsangnam-do, Republic of Korea. Hyperspectral images were taken from 13 July to 3 August 2021 between 10:00 am and 12:00 during the growing season of peach plants on sunny days. Detailed weather information can be found at Korea Meteorological Administration (https://data.kma.go.kr/). In all, 180 ‘Madoka’ samples were collected and divided into 2 subsets: a calibration set (70 % of the total samples) for model development and a validation set (30 % of total samples). After taking field-based images of the samples, they were then taken to an indoor laboratory for the performance comparison between the two different methods.

### 4.2. HSI System

Both a lab- and field-based HSI system were utilized for taking indoor and outdoor sample images. Their conceptual diagrams are shown in Figure 3. The main components of the laboratory-based system consist of a Vis/NIR hyperspectral camera (Specim IQ, Specim Ltd., Oulu, Finland), 2 650 W halogen lamps (H1000, FOMEX, Seoul, Republic of Korea), a calibration plate for relative reflectance correction (99% barium sulfate [BaSO4] reference plate), and a darkened room for sample image acquisition. For the field-based HSI application, the same hyperspectral camera was used only under sun light conditions. Then, these sample images were analyzed.

In general, the line-scan method is used for agricultural applications because food products linearly move along a production line. Moreover, online application is well suited to line scanning. However, in the current study, a hyperspectral camera for area scanning, which is able to obtain two dimensional hyperspectral images (x and y plane) with full spatial data of a target at once, was selected due to its rapid image acquisition ability for field application [43,44].

Halogen lamps are a typical broadband light source, which can perform reflectance and transmittable imaging because they produce a stable and continuous spectrum from the visible to infrared wavelength region without sudden peaks [3]. The Vis/NIR region has been widely used for HSI applications in the fields of agricultural NDE owing to its strong response to the major chemical bonds of C–H, N–H, and O–H functional groups at specific frequencies. Therefore, halogen lamps were selected as the illumination source for our analysis. In addition, as shown in Figure 3a, a dark curtain was used to prevent external light noise.

### 4.3. Spectral Calibration and Image Acquisition

The spectral range of the hyperspectral camera used was 400 nm to 1000 nm with a 2.9 nm interval (total of 204 wavebands). The region of interest (ROI) was composed of 512 × 512 pixels per single sample image. The background regions of the acquired sample images were removed by perClass Mira software because the background pixels include totally irrelevant spectral information from the sample pixels. The field of view of the camera was 200 × 200 mm at a distance of 300 mm. The mean spectra within the ROI region were then acquired to analyze both sample types.

For every hyperspectral image, the sample and white reference images were simultaneously taken for spectral (reflectance) calibration. The reflectance calibration is an important process to minimize the quantum effect of the imaging sensor mounted in the hyperspectral camera [45]. It produces an unwanted irregular radiance even when taking the same sample images under the same conditions [7]. To prevent this, a white reference and dark current images are typically captured, and the relative reflectance of a target is then calculated. In this study, white reference images were obtained without current images, because the dark current image was automatically obtained by the camera used. Spectral calibration was performed for both lab- and field-based imaging.

### 4.4. Measurement of Physicochemical Properties

After obtaining the sample images, the physicochemical properties of ‘Madoka’ peaches such as chromaticity, SSC, firmness, and TA were measured for use as a calibration set for model development. According to USDA peach grade and standards, the best peach grade should have no less than one-third of its surface showing a pink or red color [46]. To apply a red color to the digital imaging system for peach grading, CIELAB color space has been widely used [41,47,48]. Among various color parameters, parameter a* is closely related to peach grading, with the axis indicating negative values for immature green and positive values for mature red [40,48]. Therefore, in this study, the a* parameter was selected as a major indicator for peach grading and was measured using a colorimeter (CR-400, Minolta, Osaka, Japan).

The firmness and SSC are also critical quality factors which can directly influence customers when purchasing fresh peaches [49]. Firmness was measured 3 times using a Sun Rheometer with an 8 mm probe (CR-100, Sun Scientific Inc., Tokyo, Japan) at a loading rate of 2 mm/s after taking all of the lab- and field-based hyperspectral peach sample images. The measuring point of firmness was the same location where the hyperspectral images of the ‘Madoka’ samples were taken. The SSC was measured using a digital refractometer (PAL-1, Atago Co. Ltd., Tokyo, Japan) after all samples were ground to juice. The refractometer used had a refractive index accuracy of ± 0.2, and the °Brix (%) range was 0 to 53% with a 0.1 % Brix resolution at room temperature. The TA was measured using a pH meter (BP3001, Trans Instruments, Jalan Kilang Barat, Singapore). Titration was accomplished by a 1 mL pulp diluted into 80 mL of distilled water. Then, NaOH (0.1 N) was added at a rate of 1 mL/min to reach pH 8.3.

### 4.5. Model Accuracy Parameters

In this study, the model accuracy was evaluated by R², SEP, RPD, RMSECV, and bias values. R^2^ values range from zero to one and can measure how well a statistical model predicts a result. A high R^2^ value corresponds to a model that describes the variations in its data fit well [8,44]. SEP is the parameter commonly used in the HSI technique to describe the prediction ability of the developed model. Therefore, SEP can be compared to the expected accuracy to decide whether or not the developed method is acceptable. RPD value is the ratio of the standard deviation to the root mean square errors of prediction, explaining by which factor the model accuracy has been increased compared to utilizing the mean composition for total samples. In general, quantitative prediction can be achieved when the RPD value is higher than 2. RMSECV is calculated to evaluate the performance of the partial least square model. The optimal wavelength combinations for predicting physicochemical properties can be obtained by the lowest RMSECV values [36].

### 4.6. Visualization

Prior to model development, smoothing and Savitzky–Golay derivative (SGD) 1st and 2nd derivative differentiation preprocessing techniques were applied to the raw data to improve model forecasting performance by correcting and lowering the baseline shift and multiplicative scatter effects generated from electrical devices and irregular intensities of light sources [14]. As previously mentioned, after background removal of the ‘Madoka’ sample images, the spectra of the ROI were extracted, and their spectral data were used for model development. Then, an appropriate machine learning method was automatically selected by perClass Mira software for rapid and real-time image analysis [50]. The flow chart of the model development process is represented in Figure 4.

The novel benefit of HSI is that chemical mapping of a component distribution in a target material can be accomplished by various chemometric models constructed by simultaneous measurement of spatial and spectral information [4,43,44]. Thus, in this study, pixel-based chemical images based on the spatial distribution of the physicochemical components in ‘Madoka’ samples were obtained from both lab- and field-based HSI information. Spectral data in each pixel of the sample images were extracted and utilized to analyze model prediction performance. Consequently, the prediction results were shown in color, indicating different physicochemical concentrations in the sample images, and the field-based application outcome was compared with the lab-based result to verify its prediction accuracy. All data analyses for model development, preprocessing, and visualization were conducted by perClass Mira software.

## 5. Conclusions

This study investigated the field application feasibility for the NDE of physicochemical properties in ‘Madoka’ peaches using a Vis/NIR HSI system integrated with commercial HSI analysis software. Before hyperspectral image acquisition, spectral calibration was conducted by white balance reference to prevent the quantum effect on the imaging sensor of the camera. Then, the absorption spectra of the ‘Madoka’ samples were extracted from the sample images and the spectral features of both groups were analyzed.

To analyze the physicochemical properties (chromaticity (a*), SSC, firmness, and TA), a machine learning model was automatically selected by perClass Mira software. Three preprocessing methods (smoothing, SGD 1st, and SGD 2nd) were also applied to the selected model to improve the model accuracy. The performance of the lab-based HSI analysis was conducted by using the same samples used in the field-based application.

Accordingly, statistically significant R^2^ values (about 0.85) were found in the a* values and SSC content; however, the firmness and TA content exhibited much lower R^2^ values (0.5 to 0.7). There was also a major difference in R^2^ values between the calibration and validation set. Based on these results, resultant chemical distribution images of a* values and SSC were constructed by the selected model. The final outcome clearly demonstrates that the HSI camera and commercial software used have potential for the field NDE for analyzing physicochemical properties in ‘Madoka’ peaches.

## Figures and Tables

**Figure 1 plants-11-02327-f001:**
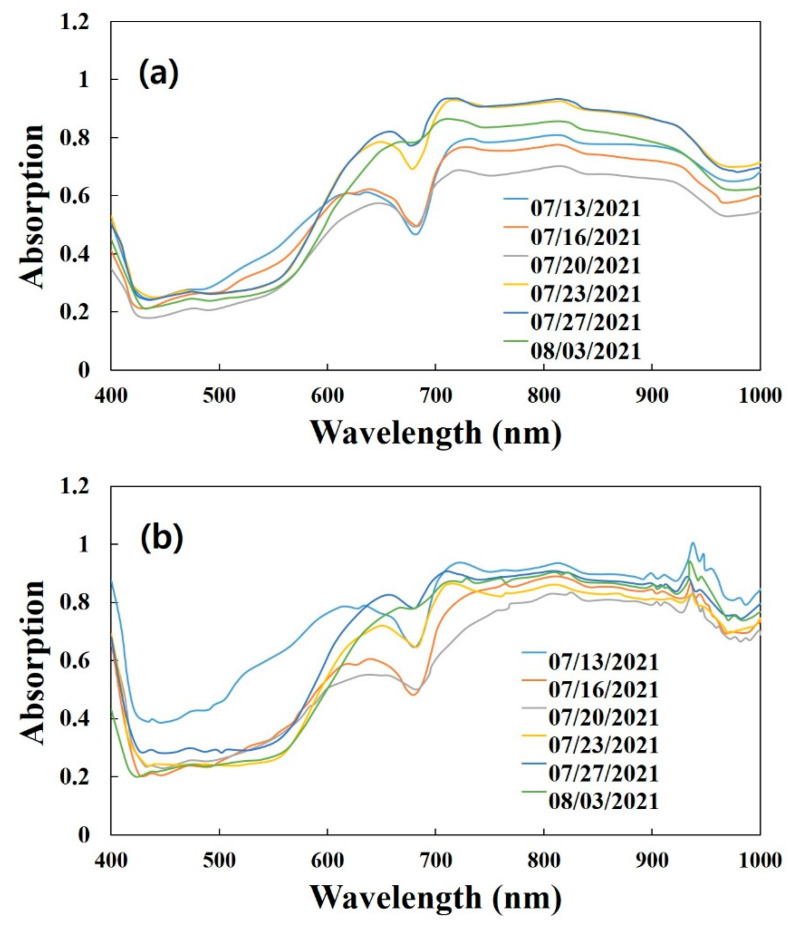
Average spectra acquired form lab- (**a**) and field-based hyperspectral images (**b**).

**Figure 2 plants-11-02327-f002:**
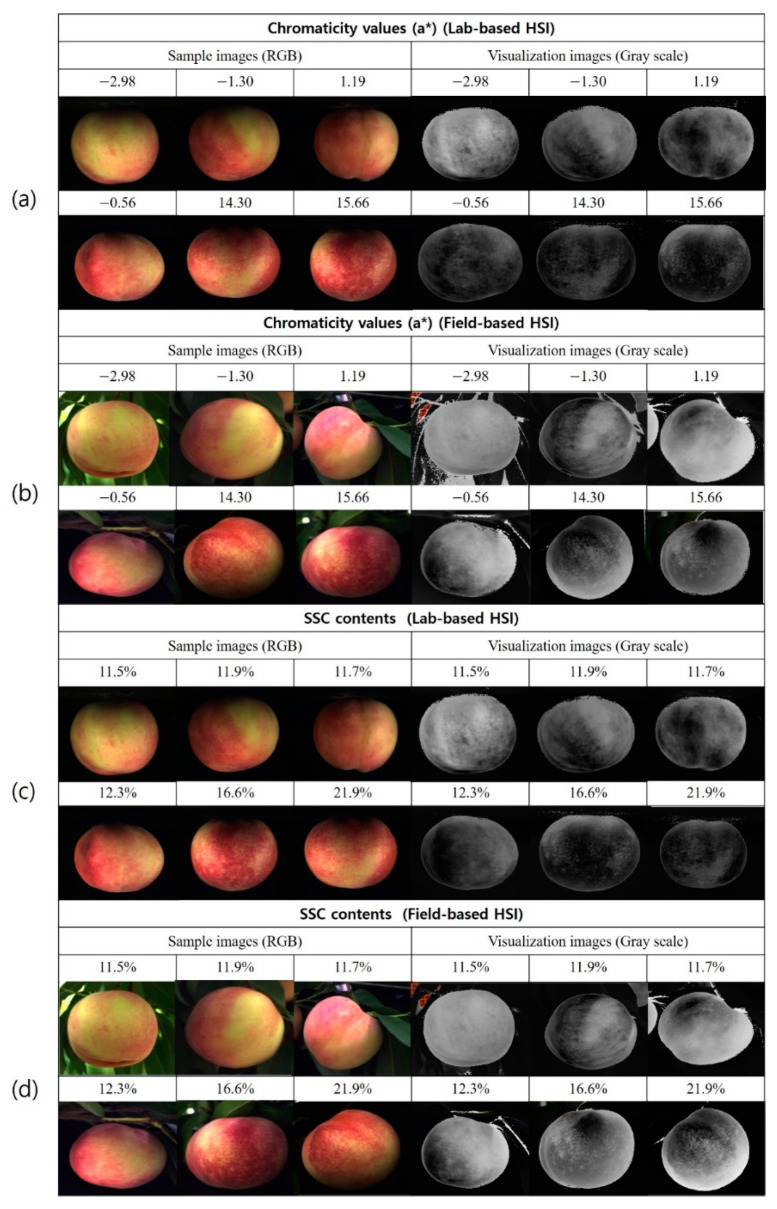
RGB and HSI images for the evaluation of chromaticity (a*) values and SSC in ‘Madoka’ peaches.

**Figure 3 plants-11-02327-f003:**
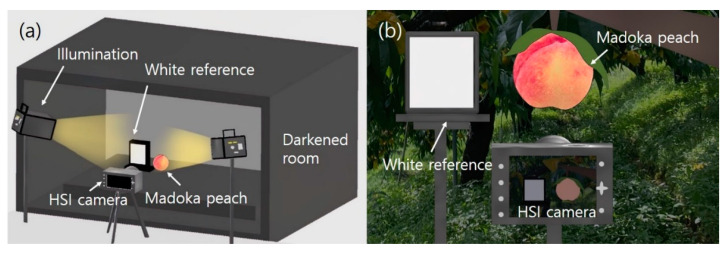
Conceptual diagram of the lab- (**a**) and field-based hyperspectral imaging system (**b**).

**Figure 4 plants-11-02327-f004:**
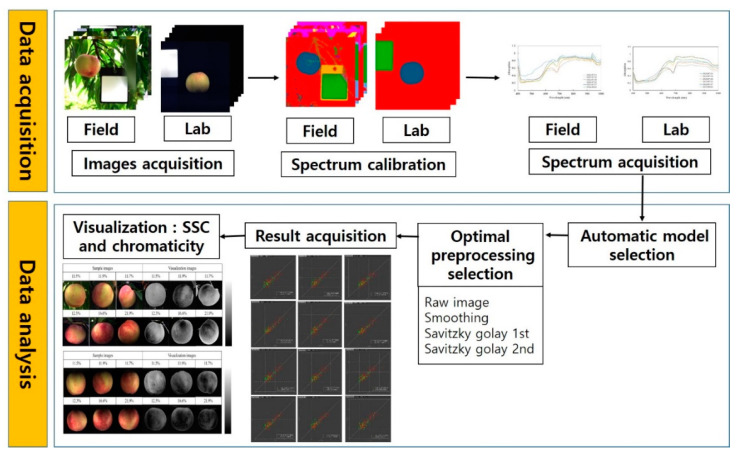
Flow chart of data processing procedures for prediction of physicochemical properties of ‘Madoka’ peaches based on lab- and field-based HSI applications.

**Table 1 plants-11-02327-t001:** Measured physicochemical properties of peach ‘Madoka’.

Parameters	Range	Average	SE	STEDV
Chromaticity (a*)	−3 to 30	12.88	1.48	8.84
SSC (°Brix)	10 to 18	13.2	0.5	2.47
Firmness (N)	25 to 50	36.77	2.18	11.88
TA (pH)	0.27 to 0.36	0.36	0.02	0.02

**Table 2 plants-11-02327-t002:** Model outcome of predicting chromaticity (a*) values of ‘Madoka’ peaches.

Place	Preprocessing	Calibration Set				Validation Set				RMSECV
		R²	SEP	RPD	Bias	R²	SEP	RPD	Bias	
Lab	Raw	0.79	3.49	2.21	−0.30	0.84	4.63	1.64	−0.92	0.02
	Smoothing	0.80	3.46	2.24	0.07	0.85	4.61	1.65	−0.27	0.01
	Savitzky golay 1st	0.89	2.57	3.01	0.11	0.87	4.21	1.81	−0.72	0.01
	Savitzky golay 2nd	0.89	2.51	3.08	−0.02	0.79	5.32	1.43	−0.67	0.01
Field	Raw	0.52	4.83	1.59	−2.23	0.82	4.27	1.81	−2.62	0.02
	Smoothing	0.89	2.54	3.03	0.01	0.85	4.59	1.68	0.65	0.01
	Savitzky golay 1st	0.85	2.93	2.62	0.10	0.84	4.59	1.68	−0.08	0.03
	Savitzky golay 2nd	0.86	2.86	2.68	−0.09	0.78	5.35	1.44	−1.11	0.02

**Table 3 plants-11-02327-t003:** Model outcome of predicting SSC concentrations of ‘Madoka’ peaches.

Place	Preprocessing	Calibration Set				Validation Set				RMSECV
		R²	SEP	RPD	Bias	R²	SEP	RPD	Bias	
Lab	Raw	0.77	1.19	2.09	0.00	0.84	1.45	1.66	0.50	0.01
	Smoothing	0.90	0.77	3.23	0.00	0.86	1.39	1.73	0.21	0.01
	Savitzky golay 1st	0.91	0.73	3.40	0.00	0.87	1.37	1.76	0.20	0.01
	Savitzky golay 2nd	0.88	0.87	2.85	0.00	0.82	1.62	1.49	0.22	0.01
Field	Raw	0.63	1.54	1.64	−0.05	0.87	1.36	1.68	−0.11	0.01
	Smoothing	0.84	1.02	2.48	0.00	0.89	1.24	1.85	0.33	0.01
	Savitzky golay 1st	0.78	1.17	2.15	0.01	0.91	1.12	2.05	0.35	0.01
	Savitzky golay 2nd	0.86	0.95	2.66	0.00	0.85	1.40	1.63	0.57	0.01

**Table 4 plants-11-02327-t004:** Model outcome of predicting firmness values of ‘Madoka’ peaches.

Place	Preprocessing	Calibration Set				Validation Set				RMSECV
		R²	SEP	RPD	Bias	R²	SEP	RPD	Bias	
Lab	Raw	0.54	8.35	1.48	0.08	0.81	7.56	1.36	−3.15	0.04
	Smoothing	0.53	8.46	1.46	0.08	0.82	7.53	1.37	−3.01	0.04
	Savitzky golay 1st	0.60	7.80	1.58	−0.08	0.82	7.72	1.33	−2.36	0.05
	Savitzky golay 2nd	0.66	7.20	1.72	0.00	0.79	8.00	1.29	−3.52	0.03
Field	Raw	0.50	8.59	1.41	0.11	0.75	9.12	1.18	−2.20	0.07
	Smoothing	0.67	7.00	1.73	0.00	0.51	11.87	0.91	−5.22	0.04
	Savitzky golay 1st	0.68	6.91	1.76	0.00	0.58	11.40	0.94	−3.72	0.06
	Savitzky golay 2nd	0.55	8.14	1.49	−0.32	0.78	8.56	1.26	−1.67	0.04

**Table 5 plants-11-02327-t005:** Model outcome of predicting TA values in ‘Madoka’ peaches.

Place	Preprocessing	Calibration Set				Validation Set				RMSECV
		R²	SEP	RPD	Bias	R²	SEP	RPD	Bias	
Lab	Raw	0.56	0.07	1.51	0.00	0.68	0.09	1.18	−0.04	0.00
	Smoothing	0.56	0.07	1.50	0.00	0.69	0.09	1.19	−0.03	0.00
	Savitzky golay 1st	0.60	0.07	1.59	0.00	0.68	0.09	1.15	−0.03	0.00
	Savitzky golay 2nd	0.71	0.06	1.86	0.00	0.67	0.09	1.13	−0.03	0.00
Field	Raw	0.73	0.06	1.92	0.00	0.63	0.10	0.90	−0.01	0.00
	Smoothing	0.56	0.07	1.51	0.00	0.75	0.08	1.11	0.01	0.00
	Savitzky golay 1st	0.78	0.05	2.14	0.00	0.74	0.08	1.07	0.00	0.00
	Savitzky golay 2nd	0.70	0.06	1.81	0.00	0.77	0.08	1.14	0.00	0.00

## Data Availability

Not applicable.

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
