# Peer review of "Field Application of a Vis/NIR Hyperspectral Imaging System for Nondestructive Evaluation of Physicochemical Properties in ‘Madoka’ Peaches"

_plants, 2022, doi:10.3390/plants11172327_

Round 1
Reviewer 1 Report
The current manuscript entitled "Field application of a visible and Near-infrared hyperspectral imaging system for nondestructive evaluation of physico-chemical properties in 'Madoka' Peaches describes the application of a hyperspectral imaging workflow to assess fruit ripening characteristics in Madoka peaches. The authors use commercially available hard- and software to assess its applicability to this particular set-up in field conditions. Hence, the novelty of this work is the application in the field. Overall, the authors show that the proposed system could work well in this particular situation. I only have minor comments.
- title is too long and should be shortened
- Page 2, 2nd sentence: grammatically incorrect. “because of its complexity. Such systems can have different scanning methods, have long image acquisition times, ….;”
- In what time frame were the sample images taken? Was it always sunny? Do the data differ during cloudy days?
- Some more information in M&M on model accuracy parameters is required (R², SEP, RPD, Bias)
- P3: table 1. For field-conditions smoothing had highest R² (0.85) but it seems SGD 1st derivative is the highest.
- What are the ranges of the measured properties. for instance, what is the range of the SSC values? This is important to get an idea about how wide the range is that can be measured with this setup. Also the sensitivity would be an issue.
Author Response
Dear journal editor and reviewers,
We are very grateful to the editor for your appropriate and constructive suggestions and for proposed correction to improve the paper quality. We also thank for the effort and time put into the review of the manuscript. The editor and reviewers have brought up some good points and we appreciate the opportunity to clarify our work objective. We tried our best to respond the concerns. The major corrections are listed below point by point. The revised texts are highlighted in red color in the revised manuscript.
- Reviewer 1
The current manuscript entitled "Field application of a visible and Near-infrared hyperspectral imaging system for nondestructive evaluation of physico-chemical properties in 'Madoka' Peaches describes the application of a hyperspectral imaging workflow to assess fruit ripening characteristics in Madoka peaches. The authors use commercially available hard- and software to assess its applicability to this particular set-up in field conditions. Hence, the novelty of this work is the application in the field. Overall, the authors show that the proposed system could work well in this particular situation. I only have minor comments.
1) Title is too long and should be shortened
The title has been changed into “Field Application of a Vis/NIR Hyperspectral Imaging System for Nondestructive Evaluation of Physicochemical Properties in ‘Madoka’ Peaches".
2) Page 2, 2nd sentence: grammatically incorrect. “because of its complexity. Such systems can have different scanning methods, have long image acquisition times, ….;”
The sentence has been changed as follows; The total cost of an HSI system is expensive, and also it is difficult for users to handle the HSI system due to its complexity, such as scanning method (line or snapshot scanning), long image acquisition time (more than regular machine vision), sensitive control for line scanning, low pixel resolution, a large amount of sample information, etc.
3) In what time frame were the sample images taken? Was it always sunny? Do the data differ during cloudy days?
Their hyperspectral images were taken between 10:00 am to 12:00 during the growing season of peach plants (July 13th to August 3rd, 2021) on sunny days. Detailed weather information can be found at Korea Meteorological Administration (https://data.kma.go.kr/). This sentence was inserted at Materials and Method section (4.1 sample preparation).
4) Some more information in M&M on model accuracy parameters is required (R², SEP, RPD, Bias)
The following paragraph has been added into ‘4.5 model accuracy parameters’ section: In this study, the model accuracy was evaluated by R², SEP, RPD, RMSECV, and bias values. R2 values range from zero to one and can measure how well a statistical model predicts a result. A high R2 value corresponds to a model that describes the variations in its data fit well [8,32]. SEP is the parameter commonly used in the HSI technique to describe the prediction ability of the developed model. Therefore, SEP can be compared to the expected accuracy to decide whether or not the developed method is acceptable. RPD value is the ratio of the standard deviation to the root mean square errors of prediction, explaining by which factor the model accuracy has been increased compared to utilizing the mean composition for total samples. In general, quantitative prediction can be achieved when the RPD value is higher than 2. RMSECV is calculated to evaluate the performance of partial least square model. The optimal wavelength combinations for predicting physicochemical properties can be obtained by the lowest RMSECV values [46].
5) P3: table 1. For field-conditions smoothing had highest R² (0.85) but it seems SGD 1st derivative is the highest.
In table 1, although the R2 value of SGD 1st derivative is higher than that of smoothing at the validation set, smoothing shows better overall prediction performance because its RPD, SEP, and bias values at both lab- and filed- applications were better.
6) What are the ranges of the measured properties. for instance, what is the range of the SSC values? This is important to get an idea about how wide the range is that can be measured with this setup. Also the sensitivity would be an issue.
Table 1 shows the measured physicochemical properties of peach ‘Madoka’. The ranges of chromaticity (a*), SSC, firmness, and TA were shown and their average, standard error (SE), standard deviation (STDEV) values were also represented, respectively. This paragraph and Table 1 have been inserted in ‘2.2 Measured physicochemical properties’ section.

Reviewer 2 Report
In this manuscript, the authors investigated the field application feasibility for NDE of physicochemical properties in ‘Madoka’ peaches using a Vis/NIR HSI system integrated with commercial HSI analysis software. The used HSI camera and commercial software have a potential for the field NDE for analyzing physicochemical properties (including a* values and SSC) in ‘Madoka’ peaches. The major concerns below need improvement:
1. What is the progress of field application of a visible and near-infrared hyperspectral imaging system for nondestructive evaluation of physicochemical properties (for example, chromaticity (a*) and SSC) in fruits? I suggest the authors expand on these in the introduction.
2. In the ‘Abstract’ section: ‘… and titratable acidity (TA) content in.’ Omit ‘in’ please!
3. In section 2.2: ‘On the other hand, under field-based HSI, smoothing has the highest accuracy (R2 = 0.85 at the validation set)’. Table 1 shows that the R2 is 0.84 (not 0.85) at the validation set. ‘smoothing’: R2 = 0.84; ‘Field Savitzky golay 1st': R2 = 0.85. Please modify!
4. In section 2.6: Omit ‘4.1 Spectral analysis’.
5. In Figure 2: Correct ‘chromaticity (a*) values (Field-based)’ to ‘chromaticity (a*) values (Lab-based)’ at the top of the chart (I guess it is in Figure 2a). And please label a, b, c, and d in Figure 2, since the figures of 2a, c, and d are mentioned in section 3.4.
6. In section 3.4: ‘This can be also observed in the SSC (see Figure 2c and d) from 12.3 to 22.9 %’. Is it 22.9% or 21.9%? Please confirm!
7. In section 4.5: Please check the position and necessity of the last two paragraphs!
8. In section 5: Modify ‘an Vis/NIR HSI system’ to ‘a Vis/NIR HSI system’. And modify ‘R2 values (about 0.85)’ to ‘R2 values (about 0.85)’.
Author Response
Dear journal editor and reviewers,
We are very grateful to the editor for your appropriate and constructive suggestions and for proposed correction to improve the paper quality. We also thank for the effort and time put into the review of the manuscript. The editor and reviewers have brought up some good points and we appreciate the opportunity to clarify our work objective. We tried our best to respond the concerns. The major corrections are listed below point by point. The revised texts are highlighted in red color in the revised manuscript.
- Reviewer 2
In this manuscript, the authors investigated the field application feasibility for NDE of physicochemical properties in ‘Madoka’ peaches using a Vis/NIR HSI system integrated with commercial HSI analysis software. The used HSI camera and commercial software have a potential for the field NDE for analyzing physicochemical properties (including a* values and SSC) in ‘Madoka’ peaches. The major concerns below need improvement:
1) What is the progress of field application of a visible and near-infrared hyperspectral imaging system for nondestructive evaluation of physicochemical properties (for example, chromaticity (a*) and SSC) in fruits? I suggest the authors expand on these in the introduction.
The following paragraph has been added to the introduction section: In the lab-based environment, critical physicochemical properties of agricultural products could be measured and predicted under a stable light source, controlled scanning speed, and minimized vibration [3,19,20]. However, the field-based HSI for agricultural products has some critical issues, such as irregular intensity of sunlight, the movement of clouds, dirt on the surface of fruits, etc. They can affect the brightness of the obtained hyperspectral images and decrease the overall HSI performance. Accordingly, improving the field-based HSI technologies' performance is an emerging issue for NDE of the agricultural product [20].
2) In the ‘Abstract’ section: ‘… and titratable acidity (TA) content in.’ Omit ‘in’ please!
'In’ has been removed in the ‘Abstract’ section.
3) In section 2.2: ‘On the other hand, under field-based HSI, smoothing has the highest accuracy (R2 = 0.85 at the validation set)’. Table 1 shows that the R2 is 0.84 (not 0.85) at the validation set. ‘smoothing’: R2 = 0.84; ‘Field Savitzky golay 1st': R2 = 0.85. Please modify!
In table 1, R2 value (0.84) of smoothing at field-based HSI has been changed into (0.85) and R2 value (0.85) of Savitzky golay 1st at field-based HSI has been revised to 0.84.
4) In section 2.6: Omit ‘4.1 Spectral analysis’.
‘4.1 Spectral analysis’ has been removed.
5) In Figure 2: Correct ‘chromaticity (a*) values (Field-based)’ to ‘chromaticity (a*) values (Lab-based)’ at the top of the chart (I guess it is in Figure 2a). And please label a, b, c, and d in Figure 2, since the figures of 2a, c, and d are mentioned in section 3.4.
‘chromaticity (a*) values (Lab-based)’has been added into the top of the chart and the label a, b, c, and d have been also inserted in Figure 2.
6) In section 3.4: ‘This can be also observed in the SSC (see Figure 2c and d) from 12.3 to 22.9 %’. Is it 22.9% or 21.9%? Please confirm!
22.9% has been revised to 21.9% in section 3.4.
7) In section 4.5: Please check the position and necessity of the last two paragraphs!
The last two paragraphs have been removed.
8) In section 5: Modify ‘an Vis/NIR HSI system’ to ‘a Vis/NIR HSI system’. And modify ‘R2 values (about 0.85)’ to ‘R2 values (about 0.85)’.
‘an Vis/NIR HSI system’ to ‘a Vis/NIR HSI system’ has been changed into ‘a Vis/NIR HSI system’ in section 5. And superscript was used to R2 values (about 0.85).
